# A Sulfur-Bridging Sulfonate-Modified Zinc(II) Phthalocyanine Nanoliposome Possessing Hybrid Type I and Type II Photoreactions with Efficient Photodynamic Anticancer Effects

**DOI:** 10.3390/molecules28052250

**Published:** 2023-02-28

**Authors:** Zixuan Chen, Yuan-Yuan Zhao, Li Li, Ziqing Li, Shuwen Fu, Yihui Xu, Bi-Yuan Zheng, Meirong Ke, Xingshu Li, Jian-Dong Huang

**Affiliations:** State Key Laboratory of Photocatalysis on Energy and Environment, Fujian Provincial Key Laboratory of Cancer Metastasis Chemoprevention and Chemotherapy, College of Chemistry, Fuzhou University, Fuzhou 350108, China

**Keywords:** photodynamic therapy, phthalocyanine, liposome, fluorescence imaging, anticancer

## Abstract

Phthalocyanines are potentially promising photosensitizers (PSs) for photodynamic therapy (PDT), but the inherent defects such as aggregation-caused quenching effects and non-specific toxicity severely hinder their further application in PDT. Herein, we synthesized two zinc(II) phthalocyanines (PcSA and PcOA) monosubstituted with a sulphonate group in the alpha position with “O bridge” and “S bridge” as bonds and prepared a liposomal nanophotosensitizer (PcSA@Lip) by thin-film hydration method to regulate the aggregation of PcSA in the aqueous solution and enhance its tumor targeting ability. PcSA@Lip exhibited highly efficient production of superoxide radical (O_2_^∙−^) and singlet oxygen (^1^O_2_) in water under light irradiation, which were 2.6-fold and 15.4-fold higher than those of free PcSA, respectively. Furthermore, PcSA@Lip was able to accumulate selectively in tumors after intravenous injection with the fluorescence intensity ratio of tumors to livers was 4.1:1. The significant tumor inhibition effects resulted in a 98% tumor inhibition rate after PcSA@Lip was injected intravenously at an ultra-low PcSA@Lip dose (0.8 nmol g^−1^ PcSA) and light dose (30 J cm^−2^). Therefore, the liposomal PcSA@Lip is a prospective nanophotosensitizer possessing hybrid type I and type II photoreactions with efficient photodynamic anticancer effects.

## 1. Introduction

Cancer continues to serve as one of the deadliest diseases, causing millions of deaths every year [1,2]. Currently, chemotherapy and radiotherapy are the two main clinical treatments, but both of them regularly cause serious systemic toxicity and adverse side effects [3,4,5,6]. Whereas, the emerging photodynamic therapy (PDT) has gained considerable interest in cancer treatment owing to various advantages including minimal trauma, favorable spatial-temporal selectivity, and non-drug resistance [7,8,9,10]. The localized light-activated photosensitizers (PSs) produce substantial reactive oxygen species (ROS) by either type I or type II mechanisms, resulting in irreversible chemical damage in the target position [8,11,12,13]. As the typical second-generation PSs, zinc(II) phthalocyanines are considered competitive candidates for PDT on account of strong absorption in the near-infrared (NIR) region, flexible modification, low dark toxicity, and strong photosensitization [14].

Zinc(II) phthalocyanines studied extensively perform treatment of diseased tissues mainly through the singlet oxygen (^1^O_2_) produced by the type II mechanism [14,15]. However, PDT via type II mechanism is regulated by oxygen concentration, and the tumor hypoxic environment typically makes zinc(II) phthalocyanines difficult to achieve the desired therapeutic effect. Unlike type II PDT, type I PDT shows great potential in overcoming hypoxic cancers of its decreased O_2_-requirement [16]. In addition to being driven by intermolecular hydrophobic forces from the conjugate macrocycle, most zinc(II) phthalocyanines tend to aggregate in an aqueous solution, resulting in fluorescence quenching and much lower ROS production [17,18]. Moreover, the poor target ability of zinc(II) phthalocyanines also seriously inhibits photodynamic anticancer efficacy [14]. Although current studies found that optimizing zinc(II) phthalocyanine structures such as introducing hydrophilic groups suppressed aggregation to some extent. Research to improve zinc(II) phthalocyanine’s targeted properties is very limited. It is still a major challenge to explore a rational delivery system for zinc(II) phthalocyanines [19,20].

In recent decades, liposomes have shown great potential as a promising nano-delivery platform for prolonging blood circulation time and enhancing the permeability of drugs into biological membranes [21,22,23,24,25]. Liposomes were originally applied to some basic studies such as biofilm photochemistry [26,27]. Liposomes perform more applications owing to their characteristics such as the effectiveness of loading both hydrophobic and hydrophilic reagents. Currently, it could be combined with PDT to deliver PSs targeted to tumor cells to accomplish their clearance without affecting healthy cells as further research [28,29,30]. Compared with inorganic materials, liposomes show better biocompatibility and biosafety [24,31,32]. Liposomal nanophotosensitizers possess enhanced permeability and retention (EPR) effects relative to small molecule PSs, promoting the preferential accumulation of PSs in tumor sites [32,33]. Several liposome formulations of anticancer medicines have been approved by the FDA, of which Visudyne^®^ was the first liposomal PS in FDA approval [28,34].

Herein, we successfully synthesized two sulfonate-modified zinc(II) phthalocyanines (PcSA and PcOA) in the alpha position with “S bridge” and “O bridge”. Interestingly, PcSA showed a significantly improved superoxide radical (O_2_^∙−^) generation compared to PcOA even in the aggregated state, suggesting that could be related to the stronger electron-giving ability of the “S” atom. Then, we used a thin-film hydration method to construct a liposomal nanophotosensitizer (PcSA@Lip). PcSA@Lip possessed hybrid type I and type II photoreactions and exhibited highly efficient production of O_2_^∙−^ and ^1^O_2_ in water under light irradiation, which were 2.6-fold and 15.4-fold higher than those of free PcSA, respectively. PcSA wrapped with liposome effectively suppressed aggregation and improved photodynamic activity and tumor selectivity in vitro/vivo. Overall, PcSA@Lip has successfully achieved in vivo fluorescent tracing and photodynamic tumor therapy in mice with enormous potential for practical and clinical applications. It is worth emphasizing that phthalocyanine-based PSs with efficient hybrid type I and type II mechanisms are still rare. This work may provide a valuable reference for the design of new PSs.

## 2. Results and Discussion

### 2.1. Synthesis and Characterization of Phthalocyanines

Phthalocyanine has already emerged as a promising second-generation PSs due to strong absorption in the visible and near-infrared regions (600–800 nm) and ease of structural modification [35,36]. It is an urgent requirement to achieve dispersibility and higher ROS generation in an aqueous solution in order to reach a broader application of phthalocyanines. Wen et al. developed type I photodynamically enhanced thiophene isoindigo derivatives by modulating sulfur group elements [37]. Studies have shown that the modification of zinc(II) phthalocyanines in the alpha position with the “O bridge” and “S bridge” as bonds causes a red shift in the absorption and fluorescence spectra because of the electron-donating ability of “O” and “S” atoms [19,38,39]. Therefore, as shown in Figure 1, we synthesized two monosubstituted sulfonate-modified zinc(II) phthalocyanines in the alpha position using “S bridge” and “O bridge” as the linkages (PcSA and PcOA), respectively. The detailed synthesis and characterization of PcSA and PcOA are shown in the Appendix A. 

The preliminary determination of the basic properties of PcSA and PcOA in dimethyl formamide (DMF) revealed the absorption and fluorescence spectra peaks of PcSA were slightly red-shift compared with those of PcOA (Figure 1a,b and Appendix A), which should be induced by the enhanced electron-donating ability of the “S” atom. The absorption peaks of both PcSA and PcOA in water turned out to be broader and weaker compared to those in DMF, which indicated that they probably form aggregates in water. In addition, their fluorescence in water was almost completely quenched. Appendix A confirmed that PcSA forms irregular aggregates in water. Methylene blue (MB) was selected as the control for this experiment because it is a water-soluble, clinically used PS that operates through both type I and type II mechanisms [40,41,42]. We investigated the ability of PcSA and PcOA to produce ROS, and the result showed O_2_^∙−^ generated by PcSA was 3.4 folds higher than that of PcOA (Figure 1c and Appendix A). This is probably because the “S” atom has a stronger electron-giving ability than the “O” atom, resulting in more O_2_^∙−^ production by PcSA [37,42,43]. Since PcSA and PcOA are aggregated in water, the ^1^O_2_ output was lower than that of MB (Figure 1d and Appendix A). PcSA and PcOA hardly produced photothermal activity in the water, according to Appendix A. Generally, PSs could release absorbed photons by fluorescence emission, or vibrational relaxation to generate heat or intersystem crossover to the lower energy triplet state (T_1_) to produce ROS through electron transfer or energy transfer. The above studies show that PcSA provides an improved type I photodynamic activity in water compared to PcOA. 

### 2.2. Preparation of PcSA@Lip

The fluorescence of PcSA in water was significantly quenched due to the aggregation-caused quenching effect of hydrophobic PcSA. The PDT efficacy of PcSA in vivo could be hindered by irregular aggregation in an aqueous solution. Therefore, we prepared PcSA@Lip by thin-film hydration method with a molar ratio of 35:22:1 of lecithin, cholesterol and PcSA (Figure 2a). The encapsulation rate and drug loading of PcSA@Lip were 85.50% and 2.08%, respectively, as calculated by the absorption spectroscopic method.

The dynamic light scattering (DLS) analysis demonstrated that PcSA@Lip exhibited a uniform and stable particle size in water, with an average particle size of 150 nm and constant mean size in an aqueous solution for more than a week (Figure 2b,c). Meanwhile, the image of the transmission electron microscope (TEM) showed that PcSA@Lip presented a uniformly spherical shape with a diameter of 120 nm (Figure 2d). From the absorption and fluorescence emission spectra of PcSA and PcSA@Lip (Figure 2e,f), the Q-band of PcSA@Lip in an aqueous solution was significantly closer to the monomeric peak than that of PcSA and a slight fluorescence could be seen, indicating that PcSA@Lip effectively inhibited the aggregation of PcSA in aqueous solution.

### 2.3. High O_2_^∙−^ and ^1^O_2_ Generation of PcSA@Lip

To investigate the photosensitive activity of PcSA@Lip, we examined the generation of O_2_^∙−^ and ^1^O_2_ in water using a dihydroethidium (DHE) probe and a singlet oxygen sensor green (SOSG) probe, respectively [42,44,45]. According to Figure 3a,b and Appendix A, PcSA@Lip displayed a significantly improved ability to produce ROS through type I and type II mechanisms compared to the known PS MB. Moreover, PcSA@Lip produced 2.6 and 15.4 times more O_2_^∙−^ and ^1^O_2_ under light irradiation than those of PcSA, respectively. PcSA@Lip showed a significant increase in the generation of O_2_^∙−^ and ^1^O_2_ in aqueous solution probably due to the aggregation inhibition and the more photons absorption. In addition, Figure 3c showed that PcSA@Lip and PcSA in water both had no significant photothermal conversion capability.

### 2.4. Highly Efficient Cellular Uptake and Phototherapeutic Properties In Vitro

The photodynamic effect of PcSA@Lip and PcSA against hepatocellular carcinoma cells HepG2 was determined using the 3-(4,5-dimethyl-2-thiazolyl)-2,5-diphenyl-2*H*-tetrazolium bromide (MTT) method[46]. As shown in Appendix A, both PcSA@Lip and PcSA showed almost no cytotoxicity in the dark environment. However, under red light irradiation (λ > 610 nm, 15 mW/cm^2^, 30 min), the cytotoxicity of PcSA@Lip was significantly higher than that of PcSA. The half maximal inhibitory concentration (IC_50_) value of PcSA@Lip is approximately 11.3-fold lower than that of PcSA (Figure 4a and Appendix A). The live/dead cell staining assays also demonstrated that PcSA@Lip had a stronger cytocidal efficacy than PcSA at the same concentration (Figure 4b). To explore the reason for the higher cellular phototoxicity of PcSA@Lip, we examined the uptake of PcSA@Lip and PcSA by HepG2 cells. As shown in Figure 4c and Appendix A, the cellular uptake of PcSA@Lip was about 11.5-fold higher than that of PcSA. The higher cellular uptake of PcSA@Lip probably promotes photodynamic efficiency in vitro.

To further investigate the intracellular ROS production by PcSA@Lip and PcSA, fluorescein 2,7-dichloroacetate (DCFH-DA) was used as a fluorescent probe to monitor ROS production in HepG2 cells incubated with PcSA@Lip and PcSA before and after light irradiation. As demonstrated in Figure 4d,e, the ROS produced by PcSA@Lip was about 12.3 times more than that of PcSA. In addition, we further used the SOSG probe and DHE probe to explore the species of ROS produced by PcSA@Lip and PcSA in HepG2 cells. Interestingly, both PcSA@Lip and PcSA could generate O_2_^∙−^ and ^1^O_2_, and there is a significant improvement of PcSA@Lip to produce ROS by type I and type II methods compared to PcSA (Figure 4f,g). These results further demonstrated that PcSA@Lip could generate ROS through type I and type II photoreaction in vitro.

### 2.5. Biodistribution and In Vivo Anticancer Effect

Firstly, we investigated the tissue distribution of PcSA@Lip and PcSA in the tumor-bearing mice before studying the tumor suppression rate. The biodistribution behavior of PcSA@Lip and PcSA in mice-bearing hepatocarcinoma (H22) tumors was studied using the IVIS Lumina III imaging system since the H22 tumor-bearing mice model was relatively accessible to establish. The mice were injected with PcSA@Lip and PcSA ([PcSA] = 200 μM, 100 μL) intravenously and the fluorescence images were continuously monitored at different time points until 24 h, respectively. It was found that PcSA@Lip started to appear at the tumor site at 1 h after injection, and the fluorescence signal gradually increased until the maximum at 8 h (Figure 5a,b). Overall, the fluorescence signal of PcSA behaved more slowly and weakly relative to PcSA@Lip, suggesting that PcSA@Lip has a stronger tumor accumulation capability than PcSA in vivo. The mice were sacrificed at 24 h post-injection, and Figure 5c,d revealed the fluorescence intensity of tumor tissue and main organs. It was noted that the fluorescence signals of PcSA@Lip and PcSA were mainly distributed in the tumor tissue, and the fluorescence signal of PcSA@Lip in the tumor tissue was 4.1-fold higher than that in the liver, significantly superior to some reported PSs [20,47,48]. This could be due to the EPR effect leading to more accumulation of PcSA@Lip at the tumor site.

Inspired by the outstanding tumor accumulation of PcSA@Lip, the photodynamic therapeutic effects of PcSA@Lip and PcSA under laser irradiation were investigated on the H22 xenograft tumor-bearing mice model (Figure 6a). The mice were divided into six groups (5 mice per group): (i) phosphate buffer saline (PBS), (ii) PBS + L (680 nm, 0.1 W/cm^2^, 5 min), (iii) PcSA, (iv) PcSA + L (680 nm, 0.1 W/cm^2^, 5 min), (v) PcSA@Lip, (vi) PcSA@Lip + L (680 nm, 0.1 W/cm^2^, 5 min) ([PcSA] = 200 μM, 100 μL). As shown in Figure 6b, the tumor of the mice treated with PcSA@Lip followed by laser irradiation showed hard growth within 14 days, with a tumor growth inhibition rate of 98%. In contrast, the tumor inhibition rate was only 56% of the group treated with PcSA followed by laser irradiation. The results in Figure 6c,d further confirmed that PcSA@Lip had a good PDT antitumor effect. In addition, no significant body weight changes were observed for all mice (Figure 6e). The above studies all indicate that PcSA@Lip is a promising nanoplatform for PDT treatment.

## 3. Materials and Methods

### 3.1. Materials and Instruments

All reactions were carried out under nitrogen atmosphere. Dimethyl sulfoxide (DMSO) and n-pentanol were dried over molecular sieves and distilled further under reduced pressure before using for reaction. Sodium 3-mercapto-1-propanesulfonate was purchased from TCI Shanghai, Shanghai, China. Sodium 3-hydroxy-1-propanesulfonate and phthalonitrile were brought from Aladdin Shanghai, China, and Acros Organics, respectively. Notably, 2,7-dichlorofluorescin diacetate (DCFH-DA), dihydroethidium (DHE), and ctDNA were purchased from Shanghai yuanye Bio-Technology Co., Ltd., Shanghai, China. Singlet oxygen sensor green (SOSG) was brought from Maokang Biology Co., Ltd., Shanghai, China. Chromatographic purifications were carried out on silica gel columns (100–200 mesh, Qingdao Haiyang Chemical Co., Ltd., Qingdao, China) using the selected eluents. All other solvents and reagents were analytical grade and employed as received. 

Fluorescence emission and absorption spectra were detected on an Edinburgh FL900/FS900 spectrofluorometer (Ediburgh Instruments Ltd., Edinburgh, UK) and Shimadzu UV-2450 UV-vis spectrometer (Shimadzu Corporation, Kyoto, Japan), respectively. High-resolution mass spectra (HRMS) were determined on a Thermo Fisher Scientific Exactive Plus Orbitrap LC/MS spectrometer (Thermo Electron Corporation, Shanghai, China). ^1^H NMR spectra were measured on a JEOL 500 spectrometer (500 MHz) or a Bruker AVANCE III 400 spectrometer (400 MHz) (BRUKER Corporation, Shanghai, China) in DMSO-d_6_ or D_2_O. Chemical shifts were relative to internal SiMe_4_ (δ = 0 ppm). Dynamic light scattering (DLS) was characterized using a Particle Analyzer (Anton Paar Litesizer 500). Transmission electron microscope (TEM) images were collected on a TECNAI G2 F20 (FEI; college of chemistry, Fuzhou University, Fuzhou, China) operating at 200 kV.

### 3.2. Synthesis of Phthalocyanines

#### 3.2.1. Synthesis of PTSA

Notably, 4-nitrophthalonitrile (1.7 g, 10.0 mmol), sodium 3-mercapto-1-propanesulfonate (1.8 g, 10.0 mmol), and K_2_CO_3_ (2.8 g, 20.0 mmol) were mixed in 10 mL DMSO and stirred at room temperature for 48 h. The reaction solution was filtered, and the filtrate was mixed with CHCl_3_ to precipitate yellow sediment. The filter residue was washed with ethanol and dried to obtain a pure white solid (2.3 g, 75%). ^1^H NMR (500 MHz, D_2_O, ppm) δ 7.91–7.84 (m, 1H, Ar-H), 7.78–7.74 (m, 2H, Ar-H), 3.29 (t, *J* = 7.0 Hz, 2H), 3.06 (t, *J* = 7.5 Hz, 2H, CH_2_), 2.11–2.05 (m, 2H, CH_2_). HRMS (ESI): *m*/*z* calcd for C_11_H_9_N_2_O_3_S_2_ [M-Na]^−^, 281.0049; found 281.0060.

#### 3.2.2. Synthesis of PcSA

A mixture of PTSA (0.3 g, 1.0 mmol), phthalonitrile (0.6 g, 5.0 mmol), K_2_CO_3_ (0.1 g, 1.0 mmol) and Zn(OAc)_2_ (0.4 g, 2.0 mmol) was heated to 100 °C in 30 mL n-pentanol, and then added 1,8-diazabicyclo [5.4.0]undec-7-ene (DBU) (0.5 mL). The reaction solution was stirred at 145 °C overnight. After a short cooling, the volatiles were removed with rotary evaporation in vacuum. The residue was purified by silica gel column chromatography using ethyl acetate (EA) and N,N-dimethylformamide (DMF) as eluent. The crude product was further purified by size exclusion chromatography using DMF as eluent to obtain compound PcSA (0.1 g, 15%). ^1^H NMR (500 MHz, DMSO-*d*_6_, ppm) δ 9.27–9.01 (m, 6H, Pc-H_α_), 8.86–8.85 (m, 1H, Pc-H_α_), 8.23–8.15 (m, 6H, Pc-H_β_), 7.98–7.91 (m, 2H, Pc-H), 3.59 (t, *J* = 8.0 Hz, 2H, CH_2_), 2.94 (t, *J* = 7.5 Hz, 2H, CH_2_), 2.41–2.35 (m, 2H, CH_2_). HRMS (ESI): *m*/*z* calcd for C_35_H_21_N_8_O_3_S_2_Zn [M-Na]^−^, 729.0464; found 792.0496.

#### 3.2.3. Synthesis of PTOA

A mixture of 4-nitrophthalonitrile (1.7 g, 10.0 mmol), sodium 3-hydroxy-1-propanesulfonate (1.6 g, 10.0 mmol), and K_2_CO_3_ (2.8 g, 20.0 mmol) was stirred in 10 mL DMSO at 100 °C for 48 h. The subsequent treatment procedure was consistent with that of PTSA to obtain the product PTOA (1.8 g, 63%). ^1^H NMR (400 MHz, D_2_O, ppm) δ 6.93–6.89 (m, 1H, Ar-H), 6.49–6.45 (m, 2H, Ar-H),2.90–2.89 (m, 2H, CH_2_), 2.51–2.47 (m, 2H, CH_2_), 1.56–1.53 (m, 2H, CH_2_). HRMS (ESI): *m*/*z* calcd for C_11_H_9_N_2_O_4_S [M-Na]^−^, 265.0278; found 265.0288.

#### 3.2.4. Synthesis of PcOA

A mixture of PTOA (0.3 g, 1.0 mmol), phthalonitrile (0.6 g, 5.0 mmol), K_2_CO_3_ (0.1 g, 1.0 mmol), and Zn(OAc)_2_ (0.4 g, 2.00 mmol) was heated to 100 °C in 30 mL n-pentanol, and then added DBU (0.5 mL). The reaction was stirred at 145 °C overnight. The purification procedure was in accordance with PcSA to obtain compound PcOA (0.09 g, 12%). ^1^H NMR (400 MHz, DMSO-*d*_6_, ppm) δ 9.55–8.99 (m, 7H, Pc-H_α_), 8.26–8.01 (m, 8H, Pc-H_β_), 3.23–3.22 (m, 2H, CH_2_), 2.18–2.17 (m, 2H, CH_2_), 1.69–1.73 (m, 2H, CH_2_). HRMS (ESI): *m*/*z* calcd for C_35_H_21_N_8_O_4_SZn [M-Na]^−^, 713.0692; found 713.0762.

### 3.3. Preparation of PcSA@Lip

PcSA@Lip was prepared using thin-film hydration method and then squeezed as previously reported [49]. The molar ratio of lecithin, cholesterol, and PcSA was 35:22:1. PcSA dissolved in DMF was added dropwise to the CHCl_3_ solution containing lecithin and cholesterol. The solution was dried by rotary evaporation under vacuum to form a thin film, and further evacuated for 1 h. Then, the thin film was sonicated with distilled water hydration for 30 min to form a liposome suspension. Then, it was sonicated at 4 °C for 15 min using an ultrasonic cell crusher and squeezed with 450 nm and 220 nm filter membranes at room temperature.

The prepared nanoliposome was characterized using dynamic light scattering (Anton Paar Litesizer 500) and transmission electron microscopy (TECNAI G2 F20) (Frequency Electronics Inc., Uniondale, NY, USA).

### 3.4. O_2_^∙−^ and ^1^O_2_ Determination

The O_2_^∙−^ and ^1^O_2_ were monitored with the dihydroethidium (DHE) probe and the Singlet Oxygen Sensor Green (SOSG) probe, respectively. PSs (PcSA@Lip, PcSA and MB, [PcSA] = 4 μM, [MB] = 4 μM) and DHE (50 μM) were dissolved in water containing 250 μg/mL ctDNA. Using water containing only DHE and ctDNA as control. Fluorescence spectra (excited at 510 nm) were recorded at different irradiated times after the mixtures were upon light irradiation (λ ≥ 610 nm, 1 mW/cm^2^). Consistent with the method used to detect O_2_^∙−^, SOSG and PSs were prepared in water at 4 μM and the aqueous solution containing only SOSG was used as control. The fluorescence spectra (excited at 488 nm) of the mixtures after being light irradiation (λ ≥ 610 nm, 1 mW/cm^2^) at different times were determined with an Edinburgh FL900/FS900 fluorescence spectrophotometer (Ediburgh Instruments Ltd., Edinburgh, UK).

### 3.5. Photothermal Detection

To measure the photothermal conversion ability of PcSA@Lip and PcSA, an aqueous solution of PSs (PcSA@Lip, PcSA, and MB, [PcSA] = 10 μM, [MB] = 10 μM, 1 mL) was continuously irradiated with a 680 nm laser at 0.5 W/cm^2^ for 10 min, and the temperature change was recorded with a near-infrared thermal imaging camera (Fluke Electronic Instrument Co., Ltd., Shanghai, China). Water was used as control.

### 3.6. Cell Culture

Human hepatocarcinoma (HepG2) cells (from ATCC, Manassas, VA, USA) were incubated in DMEM (GENVIEW) culture medium containing 10% fetal bovine serum, penicillin (50 units/mL), and streptomycin (50 mg/mL) at 37 °C under a humidified 5% CO_2_ atmosphere.

#### 3.6.1. In Vitro Photocytotoxicity

The cell phototoxicity assay was performed as previously reported [46]. Briefly, HepG2 cells (1 × 10^4^ cells per well) were hatched overnight at 37 °C under a 5% CO_2_ atmosphere in 96-well plate. Then, the cells were incubated with 100 μL of different concentrations of PSs (PcSA@Lip and PcSA) for 2 h in the dark, respectively. After removing the PSs, the cells were rinsed with PBS and re-incubated with 100 μL of culture medium, followed by illumination of 27 J/cm^2^ (λ > 610 nm, 15 mW/cm^2^, 30 min). Cell viability was determined by the colorimetric 3-(4,5-dimethyl-2-thiazolyl)-2,5-diphenyl-2H-tetrazolium bromide (MTT) assay as described previously after 24 h continued incubation without light.

#### 3.6.2. Cellular Uptake

HepG2 cells (7.5 × 10^4^ cells per dish) were inoculated at 37 °C under 5% CO_2_ and cultured overnight in confocal dishes. After 24 h the old medium was aspirated and the cells were incubated with PSs (PcSA@Lip and PcSA, [PcSA] = 4 μM) for 2 h in a dark incubator. Cells were then rinsed with PBS and observed under a Leica TCS SPE confocal microscope. The fluorescence signal of PSs was collected at 640–750 nm upon 635 nm excitation. Images were processed with SPE ROI fluorescence analysis software.

#### 3.6.3. Live/Dead Cell Staining Assays

HepG2 cells (1 × 10^4^ cells per well) were maintained overnight at 37 °C under 5% CO_2_ atmosphere in 96-well plates for cell apposition and growth. Then, cells were under different treatments: (i) incubation in medium containing PcSA@Lip ([PcSA] = 0.25 μM) for 2 h in the dark (ii) incubation in medium containing PcSA@Lip ([PcSA] = 0.25 μM) for 2 h in the dark followed by light irradiation (λ > 610 nm, 15 mW/cm^2^, 30 min) (iii) incubation in medium containing 0.25 μM PcSA for 2 h in the dark (iv) incubation in medium containing 0.25 μM PcSA for 2 h in the dark followed by light irradiation (λ > 610 nm, 15 mW/cm^2^, 30 min). HepG2 cells were stained with Calcein-AM/PI Double Stain Kit after the different treatments described above[50]. The fluorescence signal was collected from 505 to 750 nm upon being excited at 488 nm.

#### 3.6.4. Detection of Intracellular ROS, ^1^O_2_ and O_2_^∙−^

Notably, 2,7-dichlorofluorescein diacetate (DCFH-DA), SOSG, and DHE were used as fluorescent probes to detect intracellular ROS, ^1^O_2_, and O_2_^∙−^, respectively. Approximately HepG2 cells (7.5 × 10^4^ cells per dish) were seeded at 37 °C under 5% CO_2_ atmosphere and cultured overnight on confocal dishes. After removing the medium, the cells were incubated in medium containing photosensitizer (PcSA@Lip and PcSA, [PcSA] = 4 μM, 400 μL) under the same condition for 1.5 h, and then DCFH-DA (10 μM) as a ROS probe was added to co-incubated for 0.5 h. Finally, cells were rinsed with PBS. Under 488 nm excitation, fluorescence images were detected from 500 nm to 600 nm with and without light irradiation (λ > 610 nm, 15 mW/cm^2^, 5 min) using a confocal microscope (Leica). The assays for ^1^O_2_ and O_2_^∙−^ were consistent with the ROS procedure, with SOSG probe and DHE probe concentrations of 5 μM and 20 μM, respectively.

### 3.7. Animal Experiment

Hepatoma H22 cells were purchased from the China Center for Type Culture Collection (CCTCC, Wuhan, China); Female ICR mice were purchased from Wushi Animal Co., Ltd. (Beijing, China). All animal studies were carried out in compliance with guidelines of the Animal Ethics Committee of Fuzhou University (2023-SG-001), and also approved by the committee. To establish a subcutaneous tumor model, H22 cells (1 × 10^7^ cells 200 μL) were inoculated subcutaneously on the right axilla of ICR mice (20–25 g).

#### 3.7.1. In Vivo Fluorescence Imaging

PcSA@Lip and PcSA ([PcSA] = 200 μM, 100 μL) aqueous solutions were injected intravenously into mice with tumor sizes of 200 mm^3^, respectively. The fluorescence images of mice in vivo from 670 nm at different time points were captured using IVIS Lumina III imaging system (excited at 605 nm). After in vivo imaging studies, mice were euthanized at 24 h post-injection. Tumors and main organs were collected from mice at 24 h after intravenous injection and their fluorescence imaging was quantified (n = 3).

#### 3.7.2. In Vivo Photodynamic Anticancer efficacy

After the tumor volume reached 100 mm^3^, the mice were divided into six groups randomly (n = 5): (i) phosphate buffer saline (PBS), (ii) PBS followed by laser irradiation (680 nm, 0.1 W/cm^2^, 5 min), (iii) PcSA, (iv) PcSA followed by laser irradiation (680 nm, 0.1 W/cm^2^, 5 min), (v) PcSA@Lip, (vi) PcSA@Lip followed by laser irradiation (680 nm, 0.1 W/cm^2^, 5 min) ([PcSA] = 200 μM, 100 μL). The mice in the (ii) (iv) (vi) group were treated with laser irradiation (680 nm, 30 J/cm^2^) at 8 h post-intravenous injection. The tumor size was measured with calipers every other day for 14 d and calculated using the following formula: volume = (tumor length) × (tumor width)^2^ × 0.5. The relative tumor volume was calculated as V_t_/V_0_ (V_t_ and V_0_ are the tumor volume monitored at moments t and t_0_, respectively).

## 4. Conclusions

In summary, we successfully synthesized two monosubstituted sulfonate-modified zinc(II) phthalocyanines (PcSA and PcOA), among which PcSA with an “S” bridge linkage displayed superior ROS generation capability. To improve the hydrophilicity and tumor targeting, we prepared PcSA as a liposomal nanophotosensitizer (PcSA@Lip) by a facile thin film hydration method. PcSA@Lip significantly inhibited the aggregation behavior of PcSA in water, and enhanced the ROS production generated by type I and type II mechanisms. PcSA@Lip exhibited increased cellular phototoxicity with the IC_50_ value as low as 0.16 ± 0.04 μM. In addition, PcSA@Lip demonstrated significantly enhanced tumor accumulation and improved photodynamic antitumor efficacy in H22 tumor-bearing mice with a tumor inhibition rate of up to 98.0% after intravenous injection. Therefore, the liposomal PcSA@Lip is a prospective nanophotosensitizer possessing hybrid type I and type II photoreactions with efficient photodynamic anticancer effects.

## Data Availability

The data presented in this study are available on request from the corresponding author.

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
