# Peer review of "A Sulfur-Bridging Sulfonate-Modified Zinc(II) Phthalocyanine Nanoliposome Possessing Hybrid Type I and Type II Photoreactions with Efficient Photodynamic Anticancer Effects"

_molecules, 2023, doi:10.3390/molecules28052250_

Round 1

Reviewer 1 Report

The manuscript is based on relatively simple syntheses of two uncomplicated phthalocyanines, so the contribution to Pcs chemistry is not very significant. However, interesting results were obtained in photophysical and especially biological tests.

Authors should explain why the -SO3Na moiety was selected for the substituent. It was very probable of previously published studies, that one this group in Pc molecule will not be sufficient to prevent aggregation and it will also not ensure satisfactory water solubility. Is it essential for PcSA@Lip activity or any other substituted Pc in this system would be of similar properties?

P. 2, lines 89-91 and elsewhere: The statement that substituents bound via oxygen generally cause red shift is not quite correct. The position of the substituent is an important point. Published studies showed that substituents bound via oxygen in peripheral (beta) positions can cause the blue shift. Authors should verify their statement by comparison of both the prepared Pcs with the absorption maxima of corresponding unsubstituted analogue measured under the same conditions.

A simple table with at least values of absorption and emission maxima, as well as fluorescence and singlet oxygen quantum yields of the prepared compounds to afford standard comparison with already known phthalocyanines and generally photosensitizers should be included.

The lecithin used should be chemically specified, especially when molar ratios of liposomes forming ingredients are stated. There are several types of lecithin available.

Author Response

The manuscript is based on relatively simple syntheses of two uncomplicated phthalocyanines, so the contribution to Pcs chemistry is not very significant. However, interesting results were obtained in photophysical and especially biological tests.

1. Authors should explain why the -SO3Na moiety was selected for the substituent. It was very probable of previously published studies, that one this group in Pc molecule will not be sufficient to prevent aggregation and it will also not ensure satisfactory water solubility. Is it essential for PcSA@Lip activity or any other substituted Pc in this system would be of similar properties?

Response: Many thanks for the reviewer’s valuable comment. The data of Log Pw/o were summarized in Table S1. The Log Pw/o of PcSA and PcOA reach to -0.97 and -0.94 respectively, which are much lower than that of ZnPc (2.39). This result indicates that the -SO3Na group improves the hydrophilicity of phthalocyanine to some extent. Thanks to the hydrophobicity-hydrophilicity balance, PcSA could be encapsulated by liposome. Phthalocyanines without any sulfonate group may remain heavily aggregated in liposome, and phthalocyanines modified with two or more sulfonic acid groups could be not easily encapsulated in liposome.

2. P. 2, lines 89-91 and elsewhere: The statement that substituents bound via oxygen generally cause red shift is not quite correct. The position of the substituent is an important point. Published studies showed that substituents bound via oxygen in peripheral (beta) positions can cause the blue shift. Authors should verify their statement by comparison of both the prepared Pcs with the absorption maxima of corresponding unsubstituted analogue measured under the same conditions. 

Response: Thank you for the good comment. We have described more accurately that substituents bound via oxygen or sulfur at the peripheral (alpha) positions could cause the redshift in the revised manuscript (P. 2, lines 70, 91 and 95). Besides, we have added the photophysical/photochemical data of newly synthesized Pcs and unsubstituted ZnPc in DMF in the revised supporting information. As shown in Table S1, the absorption and emission maxima of newly synthesized PcSA and PcOA have a slight redshift compared with ZnPc.

3. A simple table with at least values of absorption and emission maxima, as well as fluorescence and singlet oxygen quantum yields of the prepared compounds to afford standard comparison with already known phthalocyanines and generally photosensitizers should be included.

Response: Thank you for your kind reminder. We have added Table S1 in the revised supporting information to show the photophysical/photochemical data of zinc(II) phthalocyanines in DMF.

4. The lecithin used should be chemically specified, especially when molar ratios of liposomes forming ingredients are stated. There are several types of lecithin available. 

Response: Thank you for your kind reminder. We apologize that our original Figure 2 did not specify the chemical structure of lecithin and cholesterol. We have modified the figure and hope that it is now clear to illustrate the chemical structure of lecithin and cholesterol. 

Reviewer 2 Report

The work seems to me very interesting from a biomedical point of view due to the very good results of using lysosomes modified with monosubstituted zinc phthalocyanine in photodynamic therapy of liver cancer. From a chemical point of view, the work is very carefully and thoroughly prepared, but unfortunately, apart from a few moments, I did not notice anything new in this part. Although I must admit that the preparation of monosubstituted lateral phthalocyanine complexes is very difficult, and this work deserves to be published with minor corrections.

Abstract.

1. Monosulfonated zinc phthalocyanine is not a new substance and it is commercially available. Also modification of metal contained phthalocyanines with 3-mercaptopropansulphonic and 3-hydroxopropansulfonic acids is known from scientific and patent literature. Should be corrected and clarified.

Results and Discussion

General suggestion: if PcOA is inactive and has not been included in biological studies, what was the purpose of presenting this compound in the paper? As I noted above, from the synthetic point of view, it is not a new compound, and its properties do not allow it to be effectively used for biomedical purposes, so I believe it is reasonable to remove it from the article, but I leave this decision to the authors.

1. Fig. 1. Usually, unsubstituted ZnPc was used as standard in ROS generation measurements. Why the authors chose MB as a reference, please indicate literature references justifying such a choice.

2. The lack of emission of compounds in aqueous solutions (due to aggregation and low solubility, I would rather call them suspensions) is also largely influenced by the effect of concentration self-quenching of fluorescence. Therefore, it would be worth checking the emissive properties of obtained compounds in water at lower concentrations. The authors use the same concentration for both free complexes and lysosomes (4 µM), although for lysosomes in this case, the actual total concentration of phthalocyanines will be much lower.

It would be pretty good to put 1H NMR and mass spectroscopy spectra in supplementary.

Author Response

The work seems to be very interesting from a biomedical point of view due to the very good results of using lysosomes modified with monosubstituted zinc phthalocyanine in photodynamic therapy of liver cancer. From a chemical point of view, the work is very carefully and thoroughly prepared, but unfortunately, apart from a few moments, I did not notice anything new in this part. Although I must admit that the preparation of monosubstituted lateral phthalocyanine complexes is very difficult, and this work deserves to be published with minor corrections.

Reply: We are very grateful for the reviewer’s positive and valuable comments. 

Abstract.

Monosulfonated zinc phthalocyanine is not a new substance and it is commercially available. Also modification of metal contained phthalocyanines with 3-mercaptopropansulphonic and 3-hydroxopropansulfonic acids is known from scientific and patent literature. Should be corrected and clarified.

Response: Many thanks for your good comment. We agree that the synthesis methods of PcSA and PcOA are known from the patent (CN 112409365) published early by our group, but we mainly focus on the exploration of PcSA delivery systems, photosensitive activities and biological applications in this manuscript. In addition, we have revised the expression to address your concerns and hope that it is now clearer. Please see page 1 of the revised manuscript, line 15, and page 2, lines 69 and 94.

Results and Discussion

General suggestion: if PcOA is inactive and has not been included in biological studies, what was the purpose of presenting this compound in the paper? As I noted above, from the synthetic point of view, it is not a new compound, and its properties do not allow it to be effectively used for biomedical purposes, so I believe it is reasonable to remove it from the article, but I leave this decision to the authors.

Response: Thank you for your comment. We decided to retain the part of explaining PcOA. The purpose of retaining PcOA is to compare the effects of oxygen bridge and sulfur bridge on the properties of phthalocyanines and to further emphasize the advantages of sulfur bridge modified phthalocyanine.

1. Fig. 1. Usually, unsubstituted ZnPc was used as standard in ROS generation measurements. Why the authors chose MB as a reference, please indicate literature references justifying such a choice.

Response: Many thanks for your reminder. Because it is a water soluble, clinically used PS that operates through both type I and type II mechanisms, methylene blue (MB) was selected as control for this experiment (Refs. 43-45).

2. The lack of emission of compounds in aqueous solutions (due to aggregation and low solubility, I would rather call them suspensions) is also largely influenced by the effect of concentration self-quenching of fluorescence. Therefore, it would be worth checking the emissive properties of obtained compounds in water at lower concentrations. The authors use the same concentration for both free complexes and lysosomes (4 µM), although for lysosomes in this case, the actual total concentration of phthalocyanines will be much lower.

Response: Thank you for the comment. Maybe it’s due to our way of expression to confuse you. The experimental data of free complexes and lysosomes in the manuscript were measured under the condition of controlling the same concentration of PcSA. The expression also has been corrected in the revised manuscript.

It would be pretty good to put 1H NMR and mass spectroscopy spectra in supplementary.

Response: According to your suggestion, we have added the characterization of PcSA and PcOA in the revised supporting information (Figure S1-S8).

Reviewer 3 Report

In this paper, two novel sulfonate-modified zinc(II) phthalocyanines (PcSA and PcOA) with “S bridge” and “O bridge” compounds were synthesized and a liposomal nanophotosensitizer (PcSA@Lip) was prepared for in vitro/vivo biological studies. PcSA@Lip shows good tumor accumulation and increased cellular phototoxicity, making it an excellent candidate for studying type I photodynamic therapy. 

Please attach the NMR and HRMS spectra to the supporting information.

Please write the full name of DBU in the synthesis part. 

Line 23, light dose is "30 J cm-2", while line 240 says "27 J/cm2". Please confir the data. 

Author Response

In this paper, two novel sulfonate-modified zinc(II) phthalocyanines (PcSA and PcOA) with “S bridge” and “O bridge” compounds were synthesized and a liposomal nanophotosensitizer (PcSA@Lip) was prepared for in vitro/vivo biological studies. PcSA@Lip shows good tumor accumulation and increased cellular phototoxicity, making it an excellent candidate for studying type I photodynamic therapy.

Reply: Many thanks for the reviewer’s valuable and insightful comments. 

1. Please attach the NMR and HRMS spectra to the supporting information.

Response: Many thanks for your good suggestion. We have updated the characterization of PcSA and PcOA in the revised supporting information (Figure S1-S8).

2. Please write the full name of DBU in the synthesis part.  

Response: Thank you for the comment. They have been corrected in the revised manuscript.

3. Line 23, light dose is "30 J cm-2", while line 240 says "27 J/cm2". Please confirm the data.

Response: Thank you for your kind reminder. 30 J cm-2 is used in mouse experiment while 27 J/cm2 is used in cell experiment. In mouse experiment, the H22 tumor-bearing mice were treated with a 680 nm laser at 0.1 W/cm2 for 5 min. In cell experiment, HepG2 cells were irradiated with a red light (λ > 610 nm) at 15 mW/cm2 for 30 min. 

Round 2

Reviewer 1 Report

After additions and corrections the manuscript could be published with still one important correction necessary. The terminology of positions on the Pc core is oposite then now written. Alpha is NON-PERIPHERAL, beta is PERIPHERAL. So the substituents in the presented compounds are alpha and NON-PERIPHERAL, see e.g. Molecules 202227(5), 1529; https://doi.org/10.3390/molecules27051529.

Author Response

After additions and corrections the manuscript could be published with still one important correction necessary. The terminology of positions on the Pc core is opposite then now written. Alpha is NON-PERIPHERAL, beta is PERIPHERAL. So the substituents in the presented compounds are alpha and NON-PERIPHERAL, see e.g. Molecules 2022, 27(5), 1529; https://doi.org/10.3390/molecules27051529.

Response: Many thanks for the reviewer’s valuable comment. We have fixed the error to address your concerns and hope that it is now clearer. Please see page 1 of the revised manuscript, line 15, and page 2, lines 69, 90 and 94.